# Voxify3D: From Mesh to Voxel Art with Palette Discretization and Semantic Guidance

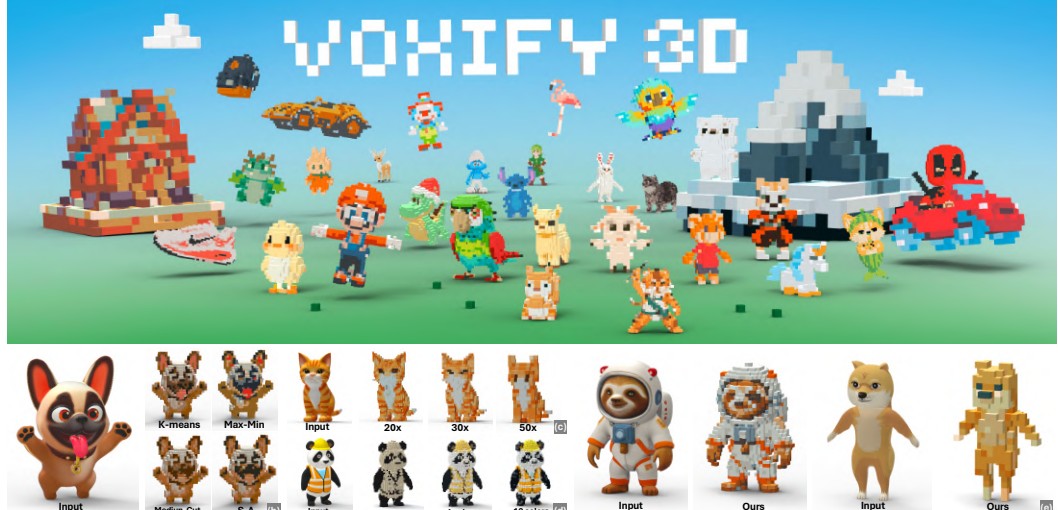

Figure 1: **Stylized voxel art with controllable abstraction.** Voxify3D converts 3D meshes into stylized voxel art using discrete color palettes, pixel art supervision, and voxel-based radiance fields. This teaser showcases the flexibility and quality of our method. (**a**) Diverse voxel art outputs across object types and use cases. (**b**) Comparison of different palette selection methods (**c**) Control over the resolution of the voxel grid ($20\times$, $30\times$, $50\times$) allows balance of detail and abstraction. (**d**) The variation in color count (2, 4, 8) shows the impact of palette size on expressiveness. (**e**) Input-output comparisons on multiple objects demonstrate faithful voxel structure, semantic clarity, and voxel art aesthetics.

## Abstract

Voxel art is a distinctive stylization widely used in games and digital media, yet creating it from 3D meshes remains labor-intensive. Existing approaches, such as downsampling or direct editing, fail to capture the abstract aesthetics and preserve essential details. We introduce **Voxify3D**, a differentiable two-stage framework for stylized voxel art generation. First, a coarse voxel grid is initialized via voxel-based 3D reconstruction. Then, the grid is refined under six-view orthographic pixel-art supervision with colors constrained to discrete palettes. Our method incorporates (1) orthographic projection with pixel art supervision to preserve sharp and essential abstract details, (2) a patch-level perceptual loss to preserve distinctive semantic features, and (3) a differentiable palette-based quantization scheme leveraging Gumbel-Softmax, which produces clear voxel renderings with distinct tonal abstraction. Experiments and user studies show that Voxify3D achieves superior visual quality, semantic fidelity, and pixel-level aesthetics compared to prior methods, providing a practical solution for automated voxel art creation.

## 1 Introduction

Voxel art is a distinctive form of 3D digital artwork, characterized by its minimalist aesthetic and discrete volumetric structure. Despite its growing popularity in games and digital media, creating

Figure 2: **Existing methods often miss key features in voxelization.** While IN2N (Haque et al., 2023), Vox-E (Sella et al., 2023), and Blender (Geometry Nodes) generate outputs that are coarse, blurry, or semantically inconsistent, they frequently lose critical elements such as facial features. In contrast, our method preserves structural details and produces visually appealing voxel art with sharp abstraction. This highlights the need for a dedicated framework that balances stylization and fidelity.

high-quality voxel art remains a challenging task that requires significant artistic expertise and manual effort. While recent works have achieved promising results in 2D pixel art stylization and sprite generation (Wu et al., 2022; Han et al., 2018b; Binninger & Sorkine-Hornung, 2024; Coutinho & Chaimowicz, 2022; Serpa & Rodrigues, 2019), these techniques do not trivially extend to 3D voxel art. Directly using 2D pixel art as input for 3D reconstruction leads to several challenges, including grid misalignment due to projection incompatibility, multi-view inconsistencies, structural distortion, and ambiguous dominant color representations.

Current voxel art generation methods from 3D meshes are limited. Simple downsampling often loses semantic features, resulting in overly coarse outputs. Although voxel-based neural radiance fields can efficiently capture 3D scenes (Sun et al., 2022; Chen et al., 2022; Fridovich-Keil et al., 2022), they are primarily designed for photorealistic rendering rather than stylistic abstraction, making them ill-suited for voxel art. Existing neural editing approaches, such as text-guided 3D editing (Haque et al., 2023; Liu et al., 2022; Wang et al., 2023b) are designed for photorealistic appearance manipulation and often struggle to produce clean, abstract voxel representations. Procedural tools such as Blender's Geometry Nodes offer voxelization pipelines, but require extensive manual tuning and offer no unified optimization framework. These tools also lack discrete color control and the ability to preserve abstract semantic features, both of which are core to the aesthetics and expressiveness of voxel art. As shown in Fig. 2, our method preserves semantic details while maintaining a coherent artistic style.

To address these challenges, we present **Voxify3D**, a method for automatically generating semantically meaningful and stylized voxel art from 3D meshes. Our approach introduces a two-stage differentiable optimization framework guided by six-view pixel art supervision, enabling abstract feature preservation and discrete palette-based color control. The first stage performs coarse-level optimization to initialize voxel geometry and color via neural volume rendering. In the second stage, we refine the voxel grid using orthographic supervision from six-view pixel art, incorporating a semantic perceptual loss to preserve meaningful features and a differentiable discrete color selection mechanism for palette-based abstraction.

Our main contributions include:

- A two-stage framework that combines voxel-based representation with orthographic supervision from six-view pixel art, enabling consistent and semantically meaningful stylization.
- A patch-level semantic perceptual loss that preserves essential features across different voxel resolutions.
- A differentiable color quantization mechanism based on Gumbel-Softmax and discrete palettes, producing sharp, abstract, and expressive voxel appearances.
- Flexible control over resolution, palette size, and color selection strategies, supporting diverse abstraction levels and visual styles.

## 2 RELATED WORK

**Pixelization Techniques.** Early methods used nearest-neighbor or bicubic interpolation (Gerstner et al., 2013), content-aware downscaling (Choi & Kim, 2015), and perceptual losses (Johnson et al., 2016). Image translation advanced with pix2pix (Isola et al., 2017), unpaired approaches (Wu et al., 2022), unsupervised variants (Han et al., 2018a), GAN-based generation (Coutinho & Chaimowicz,

2022), ML-assisted pipelines (Serpa & Rodrigues, 2019), diffusion methods (SD-$\pi$XL (Binninger & Sorkine-Hornung, 2024)), and vector-oriented adaptations (Jain et al., 2022; Xing et al., 2024; Igarashi & Igarashi, 2022). We adopt (Wu et al., 2022) for pixel art supervision in Stage 2.

**Voxel-based Representations.** NeRF (Mildenhall et al., 2021) offers high-quality reconstruction but is computationally intensive. Accelerated methods include DVGO (Sun et al., 2022), Plenoxels (Fridovich-Keil et al., 2022), TensoRF (Chen et al., 2022), Instant-NGP (Müller et al., 2022) (Garbin et al., 2021; Reiser et al., 2021; Schwarz et al., 2022), differentiable voxelization (Luo et al., 2024), unified frameworks (Wu et al., 2024b), hierarchical representations (Ren et al., 2024), sparse architectures (Chen et al., 2023b), and compression techniques (Li et al., 2023a). Voxels are common in geometry processing (Coeurjolly et al., 2018), volumetric storage (Museth, 2013), and simulations (Losasso et al., 2004). Our method employs explicit voxel storage for RGB, density, and palette-logits.

**Neural 3D Stylization.** DreamFusion (Poole et al., 2022) introduced SDS for 2D-supervised 3D generation. Text2Mesh (Michel et al., 2022) pioneered CLIP-guided abstraction, StyleRF (Liu et al., 2023) demonstrated zero-shot transfer on grids. Recent work includes NeRF-Art (Wang et al., 2023a), 3DStyleGLIP (Chung et al., 2024), Ref-NPR (Zhang et al., 2023), Style-NeRF2NeRF (Fujiwara et al., 2024), Magic3D (Lin et al., 2023) with 8× resolution improvement, and Fantasia3D (Chen et al., 2023a) disentangling geometry/appearance.

**Mesh Generation.** Recent methods leverage diffusion and sparse-view priors (Wang et al., 2018; Liu et al., 2024a; Bala et al., 2024; Lin et al., 2023; Xu et al., 2024; Hong et al., 2023; Wang et al., 2024; Huang et al., 2025; Xiang et al., 2024). Datasets like Rodin (Wang et al., 2022b) and Unique3D (Wu et al., 2024a) provide character-centric meshes.

**Multi-view Consistency.** Carve3D (Xie et al., 2024) introduces MRC metrics with RL finetuning, ConsistNet (Yang et al., 2024) enforces consistency via view/ray aggregation, SV3D (Voleti et al., 2024) uses latent video diffusion. These inform our orthographic supervision strategy.

**CLIP-based Supervision.** CLIP (Radford et al., 2021) enables image-text alignment for stylized synthesis (Frans et al., 2022), neural rendering (Wang et al., 2022a; Tang et al., 2023), image editing (Li et al., 2023b; Kim et al., 2022), and abstraction (Mokady et al., 2022; Patashnik et al., 2021). CLIP-Driven 3D Scene Graphs (Chen et al., 2024) introduces cross-modal losses for semantic decomposition.

**Differentiable Discrete Selection.** Gumbel-Softmax (Jang et al., 2017; Maddison et al., 2017) enables differentiable categorical sampling. Improvements include Decoupled Straight-Through (Shah et al., 2024) addressing temperature sensitivity. Applications span neural architecture search (Liu et al., 2019; Cai et al., 2019), discrete modeling (van den Oord et al., 2017), HQ-VAE (Takida et al., 2023) for hierarchical quantization, Content-Aware Radiance Fields (Liu et al., 2024b), and Q-DiT (Chen et al., 2025). SD-$\pi$XL (Binninger & Sorkine-Hornung, 2024) applies it for palette selection. Alternatives include VQGAN (Esser et al., 2021) and latent upscaling (Menon et al., 2020). We use Gumbel-Softmax for discrete color selection enabling coherent palette abstraction.

**Physical Fabrication.** Voxels map naturally to physical systems. Recent work includes LEGO generation (Pun et al., 2025) with physics-aware pipelines, progressive reconstruction (Ge et al., 2024), Google Earth voxelization (Lewis, 2024) (SIGGRAPH 2024 Best in Show), and voxel printing (Bader et al., 2018), motivating our LEGO-style demonstrations.

## 3 METHOD

We propose a two-stage framework for converting 3D meshes into stylized voxel art with high fidelity and semantic consistency (Fig. 3). Stage 1 (Sec. 3.1) builds a coarse voxel radiance field using DVGO (Sun et al., 2022) to establish geometric and color foundations. Stage 2 (Sec. 3.2) refines the grid under orthographic pixel-art supervision, with CLIP-based loss (Sec. 3.3) for semantic alignment and depth loss for geometric preservation. To achieve clean abstraction and a coherent palette, we replace the RGB grid with a learned color-logit grid and apply Gumbel-Softmax for differentiable palette quantization (Sec. 3.4). This pipeline retains abstract details, enforces a dominant palette, and conveys the distinctive style of voxel art across resolutions.

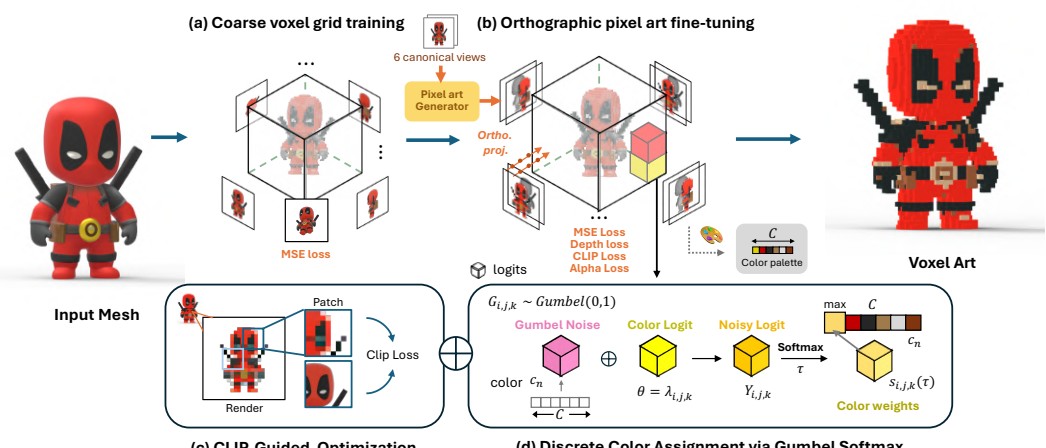

Figure 3: **Our two-stage voxel art generation pipeline.** (a) *Coarse voxel grid training:* Given a 3D mesh, we render multi-view images and optimize a voxel-based radiance field (DVGO (Sun et al., 2022)) using MSE loss to learn coarse rgb and density. (b) *Orthographic pixel art fine-tuning:* We refine the voxel grid using six orthographic pixel art views, which also serve to extract a discrete color palette (e.g., via k-means). Optimization includes appearance, depth, and alpha losses. (c) *CLIP-guided optimization:* A CLIP loss computed over rendered patches and mesh images encourages semantic alignment while being memory-efficient. (d) *Differentiable discrete color selection via Gumbel-Softmax:* Each voxel stores palette logits. Gumbel-Softmax enables differentiable sampling for end-to-end color optimization, yielding coherent, stylized voxel art.

## 3.1 COARSE VOXEL GRID TRAINING

The first stage adapts DVGO (Sun et al., 2022) to build a coarse voxel representation. Unlike NeRFs using MLPs, DVGO directly optimizes two explicit voxel grids: a density grid $d$ for spatial occupancy and a color grid $\mathbf{c} = (r, g, b)$ for appearance. This explicit structure enables faster training and efficient rendering.

We partition the object's bounding box into a grid of resolution $(W/\texttt{cell\_size})^3$, where $W$ is the canonical orthographic image width (pixels) and $\texttt{cell\_size}$ is the number of pixels per voxel edge. Each voxel stores density $d$ and RGB color $\mathbf{c}$. The rendered color $C(\mathbf{r})$ along a camera ray $\mathbf{r}$ is computed as:

$$ C(\mathbf{r}) = \sum_{k=1}^{N} T_k \alpha_k \mathbf{c}_k, \quad T_k = \exp\left(-\sum_{j=1}^{k-1} d_j \delta_j\right), \quad \alpha_k = 1 - \exp(-d_k \delta_k). \quad (1) $$

where $N$ is the number of samples along the ray, $d_k$ the density, $\delta_k$ the distance between consecutive samples, $T_k$ the accumulated transmittance, and $\alpha_k$ the opacity at sample $k$.

The coarse voxel grid is optimized with:

$$ \mathcal{L}_{\text{total}} = \mathcal{L}_{\text{render}} + \lambda_d \mathcal{L}_{\text{density}} + \lambda_b \mathcal{L}_{\text{bg}}, \quad (2) $$

where $\mathcal{L}_{\text{render}}$ minimizes the MSE between rendered and target colors to ensure visual fidelity, $\mathcal{L}_{\text{density}}$ regularizes the density to suppress noise, prevent near-clip artifacts, and employs total variation (TV) regularization to enforce spatial smoothness, and $\mathcal{L}_{\text{bg}}$ uses entropy loss to maintain clear geometry and reduce background artifacts. This stage provides a good initialization for color and density.

## 3.2 ORTHOGRAPHIC PIXEL ART FINE-TUNING

To utilize the abstract features and clean edges of pixel art for 3D grid supervision, we fine-tune the voxel space by rendering orthographic projections from six axis-aligned views and comparing them against pixel art supervision generated by the pixel art generator (Wu et al., 2022). This six-view

setup compactly covers the major surfaces of the object, while orthographic rendering formulates parallel ray casting $\mathbf{r}_i(t) = \mathbf{o}_i + t\mathbf{d}$, where $\mathbf{o}_i$ is the ray origin of pixel $\mathbf{p}_i$ and $\mathbf{d}$ is the fixed ray direction. All rays are parallel, ensuring pixel-to-voxel alignment without perspective distortions (Fig. 4).

We apply two foundational losses to supervise geometry and structure:

$$\mathcal{L}_{\text{pixel}} = \|C(\mathbf{r}) - C_{\text{pixel}}\|_2^2 \qquad (3)$$

$$\mathcal{L}_{\text{depth}} = \|D(\mathbf{r}) - D_{\text{gt}}\|_1 \qquad (4)$$

where $C(\mathbf{r})$ and $D(\mathbf{r})$ are the rendered color and depth along ray $\mathbf{r}$, $C_{\text{pixel}}$ is the RGB color from the pixel-art supervision, and $D_{\text{gt}}$ is the mesh-projected depth.

We also use an alpha loss to suppress density in background regions, enforcing background transparency to avoid floating density artifacts:

$$\mathcal{L}_{\alpha} = \|\mathcal{M}_{\alpha} \odot \bar{\alpha}\|^2, \qquad (5)$$

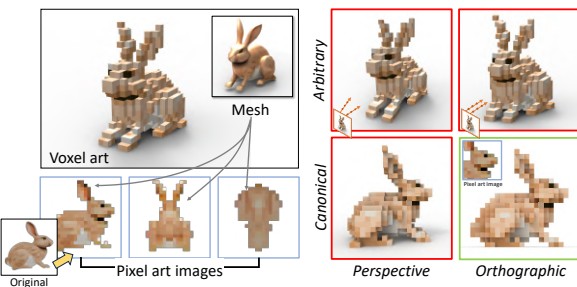

Figure 4: **Perspective vs. Orthographic.** (*Left*) Six-view pixel art pipeline. (*Right*) Perspective views (red) misalign pixels, while six orthographic views (green) enable precise pixel–voxel alignment.

where $\mathcal{M}_{\alpha} \in \{0, 1\}^{H \times W}$ is a binary mask from the pixel art alpha channel (1 for background), and $\bar{\alpha}$ denotes the accumulated ray opacity from volume rendering, which is encouraged to be $0$ for background rays to allow full transparency. This encourages transparent regions in the pixel art to remain fully transmissive, preventing the formation of undesired voxels in areas without valid supervision.

By leveraging pixel art as the supervision signal, each voxel grid more effectively captures and expresses the most important structural and appearance information.

### 3.3 CLIP-BASED SEMANTIC LOSS

To incorporate semantic supervision, we sample half of the total rays to form patches for computing a CLIP-based perceptual loss. During training, we randomly sample patch rays $(\mathbf{o}_{\text{patch}}, \mathbf{d}_{\text{patch}})$ from rendered ground-truth mesh images $I_{\text{mesh}}$. Given the rendered patch $\hat{I}_{\text{patch}}$ and the corresponding mesh-based patch $I_{\text{patch}}^{\text{mesh}}$, we extract CLIP features (Radford et al., 2021; Frans et al., 2022) and compute a perceptual loss via cosine similarity:

$$\mathcal{L}_{\text{clip}} = 1 - \cos\left(\text{CLIP}(\hat{I}_{\text{patch}}),\ \text{CLIP}(I_{\text{patch}}^{\text{mesh}})\right), \qquad (6)$$

where cosine similarity is defined as $\cos(a, b) = \frac{\langle a, b \rangle}{\|a\|\,\|b\|}$, and $CLIP(\cdot)$ denotes the CLIP image encoder output. This loss encourages voxel-rendered outputs to remain semantically aligned with the input mesh while supporting stylized abstraction, as illustrated in stage (c) of Fig. 3.

### 3.4 DISCRETE COLOR SELECTION VIA GUMBEL-SOFTMAX

To generate clean and stylized voxel appearances, while allowing flexible color selection strategies, we adopt a palette-based quantization scheme where each voxel selects a color from a predefined palette. This palette is extracted from the six-view pixel art images using a chosen clustering method before Gumbel-Softmax quantization.

Instead of regressing RGB values, each voxel $(i, j, k)$ stores a color logit vector $\boldsymbol{\lambda}_{i,j,k} \in \mathbb{R}^C$, where $C$ is the number of discrete colors in the predefined palette.

During training, Gumbel noise $\mathbf{G}_{i,j,k} \sim \text{Gumbel}(0, 1) \in \mathbb{R}^C$ is added to produce noisy logits:

$$\mathbf{Y}_{i,j,k} = \boldsymbol{\lambda}_{i,j,k} + \mathbf{G}_{i,j,k}, \qquad (7)$$

where $Y_{i,j,k,n}$ denotes the noisy logit for the $n$-th palette color at voxel $(i,j,k)$, with $n \in \{1,\dots,C\}$. A temperature-controlled softmax (Jang et al., 2017; Maddison et al., 2017) is then applied:

$$s_{i,j,k,n}(\tau) = \frac{\exp(Y_{i,j,k,n}/\tau)}{\sum_{n'=1}^{C} \exp(Y_{i,j,k,n'}/\tau)}, \tag{8}$$

where $s_{i,j,k,n}(\tau)$ is the probability of selecting the $n$-th color in the palette for voxel $(i,j,k)$, and $\tau$ is the temperature parameter controlling distribution sharpness.

In early training, we use the soft distribution $s_{i,j,k}$ directly. Later, we switch to the straight-through variant, where the forward pass uses a one-hot selection at $\arg\max_n s_{i,j,k}$, while gradients are backpropagated through the soft weights. We anneal the temperature $\tau$ during training to encourage smooth exploration in the early stages and sharper, more discrete selections later. The sampled RGB value is computed as:

$$\text{RGB}_{i,j,k} = \sum_{n=1}^{C} s_{i,j,k,n} \cdot \mathbf{c}_n, \tag{9}$$

where $\mathbf{c}_n \in \mathbb{R}^3$ is the $n$-th color in the palette.

After training, we directly select the color with the highest logit:

$$\text{RGB}_{i,j,k}^{\text{voxel}} = \mathbf{c}_{\arg\max_n \lambda_{i,j,k,n}}, \tag{10}$$

producing fully discrete voxel outputs. This process is illustrated in stage (d) of Fig. 3.

To enhance flexibility in stylization, this strategy allows users to choose the color selection method and number of colors, enabling explicit control over both color richness and overall style of the voxel art, making the design process more aligned with practical usage scenarios.

## 3.5 LOSS SUMMARY AND TRAINING PROCEDURE

The overall loss optimized during fine-tuning is a weighted sum of multiple components that jointly supervise pixel-art faithfulness, geometry consistency, semantic alignment, and spatial regularity:

$$\mathcal{L}_{\text{total}} = \lambda_{\text{pixel}} \cdot \mathcal{L}_{\text{pixel}} + \lambda_{\text{depth}} \cdot \mathcal{L}_{\text{depth}} + \lambda_{\text{alpha}} \cdot \mathcal{L}_{\text{alpha}} + \lambda_{\text{clip}} \cdot \mathcal{L}_{\text{clip}}, \tag{11}$$

where $\mathcal{L}_{\text{pixel}}$, $\mathcal{L}_{\text{depth}}$, and $\mathcal{L}_{\text{clip}}$ encourage pixel-level accuracy, depth consistency, and semantic alignment, respectively, while $\mathcal{L}_{\text{alpha}}$ suppresses background opacity to yield clean silhouettes. In Stage 2, rays are split into two groups: (1) $\mathcal{L}_{\text{pixel}}$, $\mathcal{L}_{\text{depth}}$, and $\mathcal{L}_{\text{alpha}}$, and (2) $\mathcal{L}_{\text{clip}}$ on rendered patches, all computed via volumetric rendering (equation 1). Thus, geometric supervision of the density grid is provided by $\mathcal{L}_{\text{pixel}}$, $\mathcal{L}_{\text{depth}}$, and $\mathcal{L}_{\text{alpha}}$, while semantic supervision comes from $\mathcal{L}_{\text{clip}}$, which guides voxel appearance toward the intended pixel-art style.

## 4 EXPERIMENTS

### 4.1 EXPERIMENTAL SETUP

**Dataset.** We evaluate our method on three mesh datasets: **Rodin** (Wang et al., 2022b), **Unique3D** (Wu et al., 2024a), and **TRELLIS** (Xiang et al., 2024). Rodin and Unique3D primarily feature character 3D assets with rich semantic details, making them ideal for evaluating voxel abstraction and stylized representation.

**Implementation details.** Training follows a two-stage schedule: **(a) Coarse Voxelization**: optimize the voxel grid for 8000 iterations to capture global structure; **(b) Pixel Art Supervision**: fine-tune for 6500 iterations with MSE, Depth, and CLIP losses on six orthographic views, using fixed $80 \times 80$ patches randomly sampled each iteration for CLIP loss. In the final 2000 iterations, supervision is applied only to the front view to enhance key abstract features. Gumbel-Softmax sampling is performed over a fixed palette, with temperature $\tau$ annealed from 1.0 to 0.1.

Figure 5: **Qualitative comparisons on character models from the Rodin (Wang et al., 2022b) dataset.** We compare our voxel art results with Pixel art to 3D extension, IN2N (Haque et al., 2023), Vox-E (Sella et al., 2023), and Blender's voxelization. Our method produces stylized yet consistent voxel representations with pixel art aesthetics.

**Baseline methods.** We compare against:

1. **Pixel art to 3D extension**: Render the input mesh into images, stylize them into pixel art, then train the original DVGO with these pixel-art images, using the coarse voxel grid as the final output.

2. **IN2N** (Haque et al., 2023): Language-guided mesh editing with view-consistent 3D stylization.

3. **Vox-E** (Sella et al., 2023): Language-to-voxel generation prioritizing semantics over fine geometry.

4. **Blender Geometry Nodes**: Procedural mesh-to-voxel conversion, fast but without semantic or stylization control.

### 4.2 QUALITATIVE COMPARISONS

We qualitatively compare our method with Pixel art to 3D extension, IN2N, Vox-E, and Blender on eight character meshes from the evaluation datasets (Fig. 5), with an additional eight groups of comparisons provided in the appendix.

IN2N preserves coarse structure but suffers from large variations across different guidance images, often failing to produce consistent voxelized results; Vox-E yields smoother volumes yet misses the discrete, blocky style of voxel art; Blender produces clean abstraction through procedural voxelization, which is akin to simple downsampling, but requires manual tuning and lacks semantic alignment.

Our method preserves key cues (e.g., ears, eyes) with sharp edges across $25\times$–$50\times$ resolutions, achieving both expressive stylization and semantic fidelity. Additional results are provided in the appendix and the supplementary HTML.

### 4.3 QUANTITATIVE COMPARISONS

To assess stylization fidelity and semantic preservation, we adopt the CLIP-IQA framework. For each character, we use GPT-4 to generate a detailed textual description based on the original mesh images, prepended with "A voxel art of..."

Table 1: **Average CLIP-IQA scores over all 35 examples.** Best scores are **highlighted**.

| Method | Pixel | IN2N | Vox-E | Blender | Ours |
|---|---|---|---|---|---|
| CLIP-IQA | 35.53 | 23.93 | 35.02 | 36.31 | **37.12** |

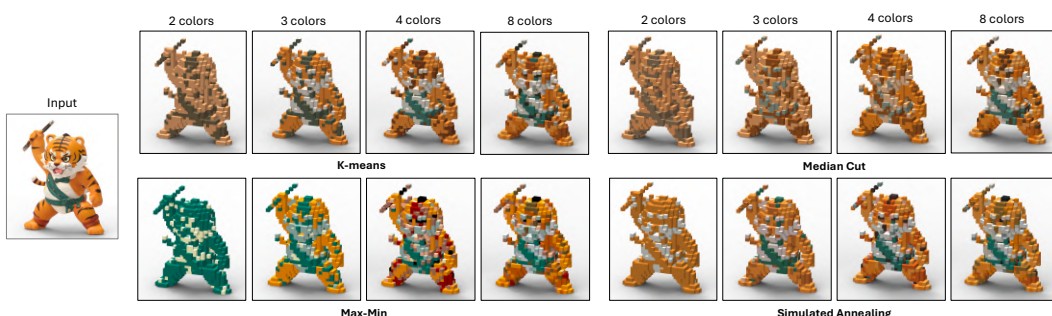

Figure 6: **Effect of Palette Selection and Color Count.** Each row corresponds to a different palette extraction method: K-means, Max-Min, Median Cut, and Simulated Annealing. Each column shows increasing color counts (2, 3, 4, 8). Each method produces unique color clustering effects.

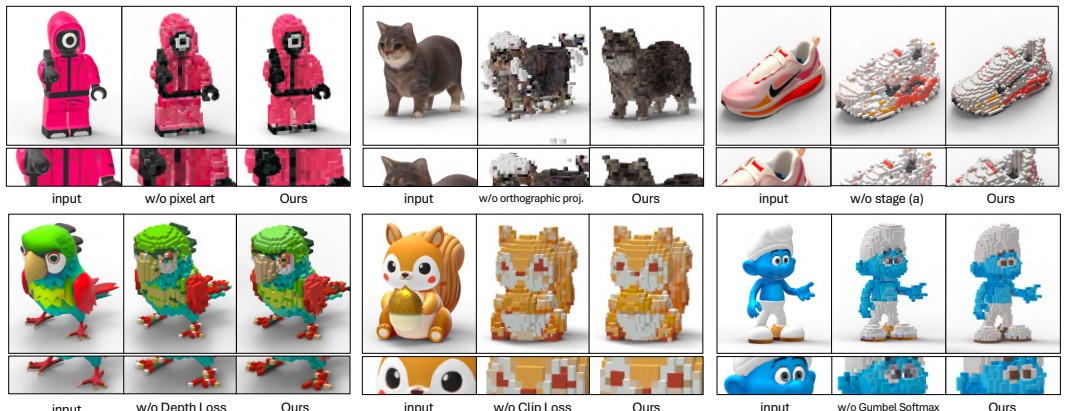

Figure 7: **Ablation study on model components.** We show outputs after removing key modules: pixel art supervision, orthographic projection, grid initialization, depth loss, CLIP loss, and Gumbel Softmax. Each row shows a different input; columns compare ablations. The full model yields coherent stylization, while removals cause distortions, color artifacts, or semantic loss.

(e.g., "A voxel art of a pink teddy bear with a red bow and heart-shaped feet"). We use OpenAI's ViT-B/32 CLIP model and compute the average cosine similarity between each prompt and the rendered images from different methods.

As shown in Table 1, the reported CLIP-IQA scores are averaged over all **35** cases. Our method consistently achieves the highest score, demonstrating superior semantic alignment and stylized abstraction across a diverse set of character meshes.

### 4.4 COLOR PALETTE CONTROLLABILITY

We evaluate the controllability of our discrete palette by varying color counts (2, 3, 4, 8) and extraction methods (K-means, Median Cut, Max-Min, Simulated Annealing), as shown in Fig. 6. More examples with additional meshes and settings are in the appendix.

### 4.5 ABLATION STUDY

We analyze the impact of each design component by removing key modules one at a time, including pixel art supervision, orthographic projection, coarse grid initialization, depth loss, CLIP loss, and Gumbel Softmax (Fig. 7). Each removal consistently leads to degraded quality: blurred abstraction, geometric distortions, and ambiguous colors, highlighting the necessity of each element.

Table 2: **CLIP-IQA ablation across voxel sizes.** CLIP loss improves semantic alignment consistently.

| Voxel Size | 25× | 30× | 40× | 50× |
|---|---|---|---|---|
| w/o CLIP Loss | 40.89 | 40.55 | 38.92 | 38.64 |
| w/ CLIP (ours) | **41.35** | **41.03** | **40.07** | **40.14** |

We compare the effect of CLIP loss across different voxel sizes. Applying CLIP loss consistently improves semantic alignment across all tested resolutions. This confirms the role of CLIP loss in maintaining character identity under voxel abstraction.

### 4.6 USER STUDY

We conducted a user study with 72 participants to evaluate our method against four baselines: Pixel Art to 3D extension, IN2N (Haque et al., 2023), Vox-E (Sella et al., 2023), and Blender Geometry Nodes. The study included 12 questions in two parts:

**(1) Stylization Evaluation (35 examples):** Participants viewed colored input meshes alongside five voxel outputs, and selected the version with the best *abstract detail* and *voxel art appeal*.

**(2) Geometry Evaluation (4 examples):** Participants compared grayscale voxel renderings and judged which better preserved the original shape.

As shown in Table 3 (a), our method received the majority of votes across all metrics: 77.90% for abstract detail, 80.36% for visual appeal, and 96.55% for geometry faithfulness, substantially outperforming all baselines.

We further conducted an expert study on color quantization with 10 art-trained participants (Table 3 (b)), where 88.89% preferred results with Gumbel-Softmax, highlighting its effectiveness in producing voxel art with dominant tones and sharp edges. Details of the study setup and additional examples are provided in the Appendix.

Table 3: **User studies.** (a) 35 examples (72 participants). (b) Color quantization (10 art-trained).

(a) Image quality (user votes, %)

| Metric | Abstract | Appeal | Geometry |
|---|---|---|---|
| Ours | **77.90** | **80.36** | **96.55** |
| Others | 22.10 | 19.64 | 3.45 |

(b) Color quantization preference (%)

| | w/o Gumbel | w/ Gumbel |
|---|---|---|
| Preferred | 11.11 | **88.89** |

## 5 CONCLUSION

We introduce **Voxify3D**, a novel framework for transforming 3D meshes into stylized voxel art with strong semantic abstraction and structural consistency. By combining coarse voxel optimization, orthographic pixel art supervision, and palette-based color quantization, our method achieves expressive and visually appealing results across a variety of character assets. Extensive experiments and user studies confirm its advantages over existing baselines in both geometric faithfulness and artistic stylization.

In addition to digital results, we further illustrate the fabrication potential of our voxel outputs by rendering them as LEGO-style assemblies (Fig. 8), demonstrating the diverse applications of our work.

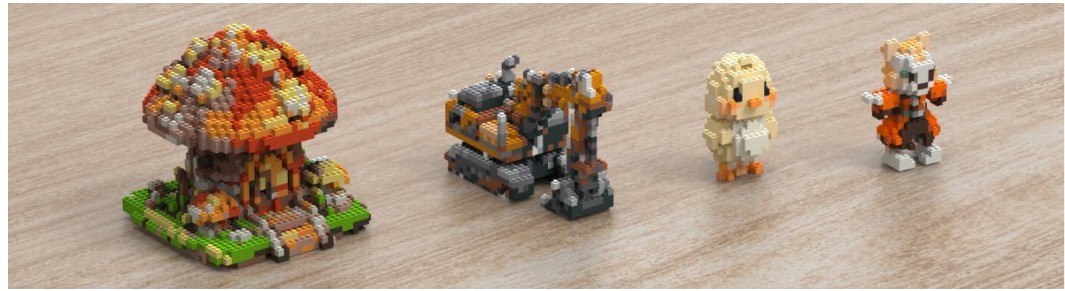

Figure 8: **Fabrication: LEGO render.** Rendered using KeyShot 2023.

**Limitations and Future Work.** Voxify3D struggles with highly intricate shapes, where thin structures or fine facial details may be lost at low voxel resolutions. Future work may explore integrating geometric priors or training strategies to enhance detail preservation and scalability, as well as adopting assembly-aware fabrication strategies inspired by LEGO brick design and connection principles to improve physical realizability of voxel-based models.

**Ethics statement.** This work focuses on voxel art stylization from 3D meshes, and does not involve human subjects, sensitive personal data, or applications with foreseeable risks of misuse. The proposed methodology is intended purely for academic and creative purposes in computer graphics and vision. We are not aware of any ethical concerns, violations of fairness, privacy, or legal compliance arising from this research.

**Reproducibility statement.** We make significant efforts to ensure reproducibility. Implementation details, hyperparameter settings, and training procedures are documented in the Appendix. Additional ablation studies and comparisons are also provided. All datasets used in the experiments are publicly available. The full source code will be released upon publication.

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

# A APPENDIX

This supplementary material provides additional details that complement our main paper. We include:

1. **Implementation Details** (Sec. A.1):
   - Codebase and training architecture.
   - Pixel art generator.
   - Logit grid initialization.
   - Parameter settings.
   - Loss design.
   - Temperature annealing schedule.
   - Cross-view inconsistency.
   - Palette selection strategies.
2. **Experimental Information** (Sec. A.2):
   - CLIP-IQA evaluation protocol.
   - User study details.
   - Expert study on color preference.
   - Run time analysis.
3. **Additional Qualitative Results and Failure Cases** (Sec. A.3):
   - More comparisons with baselines.
   - Results with varying palette settings.
   - Results under different voxel sizes.
   - Failure cases and analysis.
4. **Use of Large Language Models** (Sec. A.4).

## A.1 IMPLEMENTATION DETAILS

**Codebase and training architecture.** Our implementation builds on DVGO Sun et al. (2022). We adopt a two-stage training pipeline. In *Stage 1*, we follow DVGO to train a coarse voxel grid, which initializes both color and density representations. In *Stage 2*, the input consists of six orthographic views stylized into pixel art. Using orthographic projection, each pixel from the pixel art is directly aligned with the voxel grid, ensuring per-pixel to voxel correspondence. After 4500 iterations, training is restricted to the front view, which typically contains the most salient semantic features (e.g., facial structures), allowing the model to refine key abstract details while maintaining consistency from the earlier multi-view supervision.

**Pixel art generator.**

Our pipeline requires stylized pixel art inputs rather than simple low-resolution downsampling.

We adopt the MYOS Wu et al. (2022) generator to transform mesh renderings into high-quality pixel art, which preserves sharp boundaries and stylized abstractions. As illustrated in Fig. 9, naïve downsampling produces blurry textures, while MYOS yields pixelated structures with clear edges, better aligned with voxel abstraction.

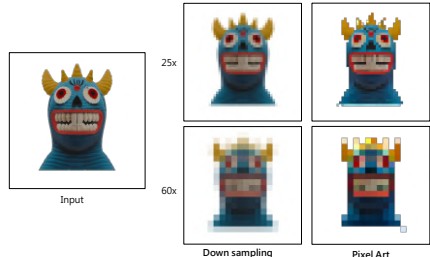

Figure 9: **Downsample vs. Pixel Art**

**Logit grid initialization.** In *Stage 2*, we initialize each voxel's logit vector by the negative distance between its *Stage 1* RGB color and the palette entries. This provides a stable bias toward closer colors and converges better than random initialization.

**Parameter settings.** We summarize the key training parameters for Stage 1 (voxel grid initialization) and Stage 2 (logit grid optimization).

**Loss design.** We adopt different objectives across the two training stages.

Table 4: **Training parameters for Stage 1 (left) and Stage 2 (right).**

| Parameter | Value | Parameter | Value |
|---|---|---|---|
| Iterations ($N_{\text{iters}}$) | 8000 | Iterations ($N_{\text{iters}}$) | 6500 |
| Batch size ($N_{\text{rand}}$) | 8192 | Batch size ($N_{\text{rand}}$) | 8192 |
| Learning rate (density grid) | $1 \times 10^{-1}$ | Learning rate (density grid) | $5 \times 10^{-3}$ |
| Learning rate (color grid $k_0$) | $1 \times 10^{-1}$ | Learning rate (logit grid) | $1 \times 10^{-1}$ |
| LR decay step | 20 | LR decay step | 20 |

Table 5: **Loss weights used in our implementation.**

| $\lambda_{\text{pixel}}$ | $\lambda_{\text{depth}}$ | $\lambda_{\text{alpha}}$ | $\lambda_{\text{clip}}$ | $\lambda_b$ | $\lambda_d$ |
|---|---|---|---|---|---|
| $\times 10$ | 10 / 20 (30 after 4500 iter) | $\times 20$ | $\times 1$ (until 6000 iter) | $\times 0.5$ (Stage 1) | $\times 0$ default (Stage 1) |

*Stage 1 (Coarse voxelization).* The voxel grid is optimized with MSE reconstruction loss, regularized by density and background terms:

$$\mathcal{L}_{\text{total}} = \mathcal{L}_{\text{render}} + \lambda_d \mathcal{L}_{\text{density}} + \lambda_b \mathcal{L}_{\text{bg}},$$

where $\mathcal{L}_{\text{render}}$ is MSE between rendered and target colors, $\mathcal{L}_{\text{density}}$ applies density regularization and total variation smoothing, and $\mathcal{L}_{\text{bg}}$ uses entropy to suppress background noise. This stage provides a stable initialization for both color and density.

*Stage 2 (Pixel-art supervision).* The fine-tuning objective combines pixel accuracy, geometry regularization, semantic alignment, and silhouette clarity:

$$\mathcal{L}_{\text{total}} = \lambda_{\text{pixel}} \mathcal{L}_{\text{pixel}} + \lambda_{\text{depth}} \mathcal{L}_{\text{depth}} + \lambda_{\text{alpha}} \mathcal{L}_{\text{alpha}} + \lambda_{\text{clip}} \mathcal{L}_{\text{clip}}.$$

**Implementation details.**

$\mathcal{L}_{\text{pixel}}$ (MSE) is up-weighted to ensure faithful color abstraction. $\mathcal{L}_{\text{depth}}$ is scaled by voxel resolution: 20 normally, and increased to 30 after step 4500. $\mathcal{L}_{\text{alpha}}$ enforces clean silhouettes via transparency regularization. $\mathcal{L}_{\text{clip}}$ is applied until step 6000, using $80 \times 80$ patches per iteration for semantic alignment. After step 6000, optimization focuses mainly on background transparency ($\mathcal{L}_{\text{alpha}}$), while CLIP loss is disabled. This scheduling ensures early semantic guidance, followed by refinement of geometry and silhouettes.

**Temperature annealing schedule.** We apply a step-wise annealing schedule for the Gumbel-Softmax temperature $\tau$, gradually lowering it to encourage sharper palette selection as training progresses. The temperature starts high to allow exploration of multiple colors, and progressively decreases to enforce deterministic palette assignments toward convergence.

**Cross-view inconsistency.** Supervision from six orthographic views keeps inconsistencies minimal, mostly near boundaries. To further refine salient cues, the last 2000 iterations are trained only on the front view (rich in facial details), reinforcing key features while preserving global consistency from earlier multi-view supervision.

**Palette selection strategies.** We explored multiple strategies for extracting compact color palettes from input images:

- *K-means clustering*: baseline method that partitions pixels into $C$ clusters and uses centroids as representative colors.
- *K-means with rare color boosting*: explicitly incorporates infrequent colors to prevent palette collapse into dominant tones.
- *Median cut*: recursively splits the RGB space by channel ranges to ensure balanced coverage of color distributions.
- *Max–min picking*: iteratively selects farthest colors in feature space to maximize palette diversity.
- *Simulated annealing*: formulates palette extraction as a discrete optimization problem, refining palettes via stochastic search.

Table 6: **Step-wise annealing schedule of the Gumbel-Softmax temperature $\tau$ during Stage 2.**

| $< 1000$ | 1000–2999 | 3000–3999 | 4000–4999 | 5000–6000 | $> 6001$ |
|---|---|---|---|---|---|
| 1.0 | 0.8 | 0.3 | 0.6 | 0.3 | 0.1 |

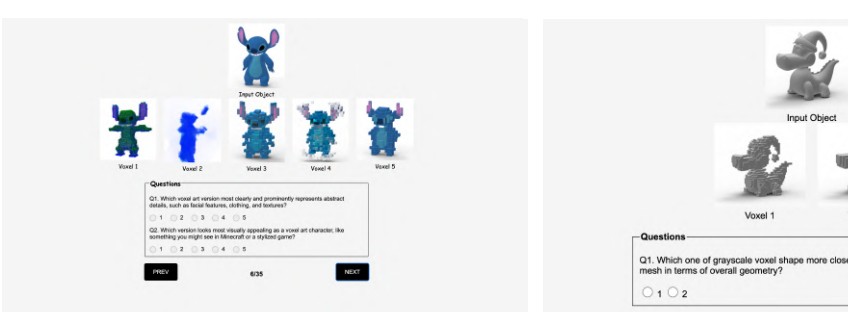

Figure 11: **User study UI.**

## A.2 Experimental Information

**CLIP-IQA evaluation protocol.** We evaluate stylization fidelity and semantic preservation using CLIP-IQA: GPT-4 generates text prompts ("A voxel art of...") from mesh images, and ViT-B/32 CLIP computes cosine similarity with rendered results, averaged over 35 cases. While training employs CLIP loss in an image–image setting, evaluation is based on GPT-4-generated text prompts. This design ensures that CLIP-IQA reflects semantic fidelity rather than overfitting to the training objective. In addition, we provide visual comparisons and a user study to further validate the reliability of the evaluation.

**User study details.**

We conducted a user study with 72 participants, who were presented with **35 colored voxel art examples** and **4 grayscale voxel renderings** Fig. 10. The interface is illustrated in Fig. 11.

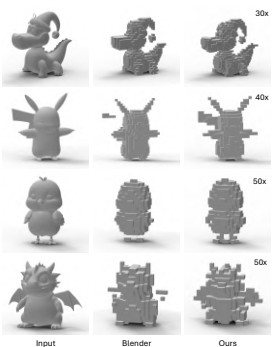

Each colored example was accompanied by the following two questions:

- **Abstract detail:** "Which voxel art version most clearly and prominently represents abstract details, such as facial features, clothing, and textures?"
- **Voxel art appeal:** "Which version looks most visually appealing as a voxel art character, like something you might see in Minecraft or a stylized game?"

For the grayscale examples, participants answered:

Figure 10: **Greyscale examples.**

- **Geometry preservation:** "Which grayscale voxel rendering more closely resembles the original 3D mesh in terms of overall geometry?"

**Expert study on color preference.** We further conducted a focused evaluation on color quantization with **10 art-trained participants**, all of whom had formal undergraduate education in art or design. Participants were asked to compare voxel art results with and without Gumbel-Softmax across **10 example pairs**, and answered the following two questions:

- **Abstract detail:** "Which voxel art version most clearly and prominently represents abstract details, such as facial features, clothing, and textures?"
- **Voxel art appeal:** "Which version looks most visually appealing as a voxel art character, like something you might see in Minecraft or a stylized game?"

As illustrated in Fig. 12, we present four representative examples comparing results *with* and *without* Gumbel-Softmax. Across responses from 10 participants on 10 question pairs, **88.89%** favored the

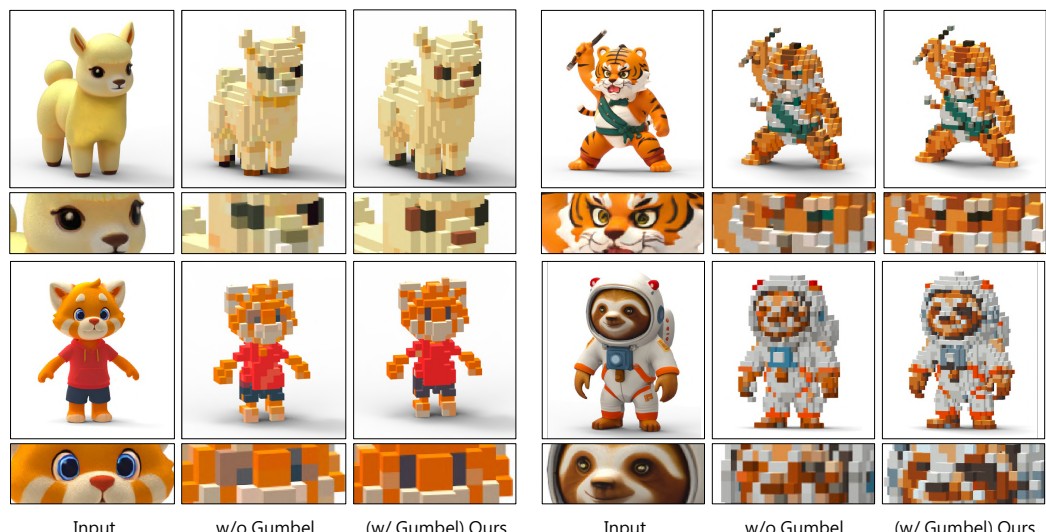

| Input | w/o Gumbel | (w/ Gumbel) Ours | Input | w/o Gumbel | (w/ Gumbel) Ours |

Figure 12: **Ablation user study of Gumbel.** Four representative examples comparing results with and without Gumbel-Softmax.

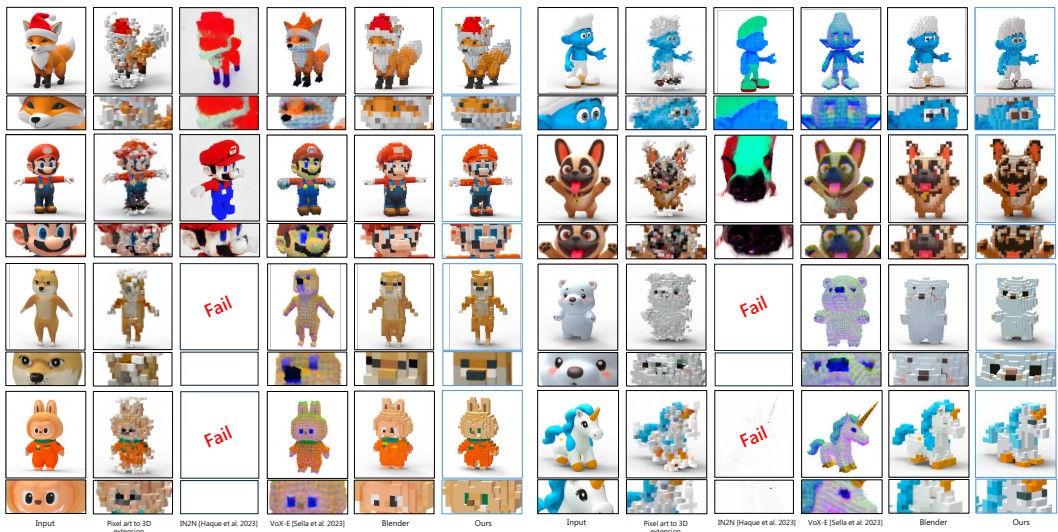

Figure 13: **Additional qualitative comparisons with baselines.** Eight representative examples compared with Pixel, IN2N, Vox-E, and Blender Geometry Nodes.

*with Gumbel-Softmax* results for voxel-art appeal, confirming its importance in producing dominant tones and clear edges.

**Runtime analysis.** On a single RTX 4090, Stage 1 (coarse voxelization) finishes in ∼8.5 minutes and Stage 2 (logit grid optimization with CLIP) in ∼108 minutes, totaling under 2 hours—substantially faster than SD-piXL (∼4h).

## A.3 ADDITIONAL QUALITATIVE RESULTS AND FAILURE CASES

**More comparisons with baselines.** In total, we evaluated **35** character models for CLIP-IQA. Here, we additionally present **8 representative examples** for qualitative comparison against the baselines: Pixel art to 3D extension, IN2N Haque et al. (2023), Vox-E Sella et al. (2023), and Blender Geometry Nodes, as illustrated in Fig. 13. While IN2N Haque et al. (2023) is effective in certain cases, we found it often fails in our setting. This is mainly because each guidance image used during training can differ significantly, leading to large inconsistencies across views.

**Results with varying palette settings.** As shown in Fig. 14, we present comparisons under different color selection strategies and palette sizes, with K-means adopted as our default palette extraction method.

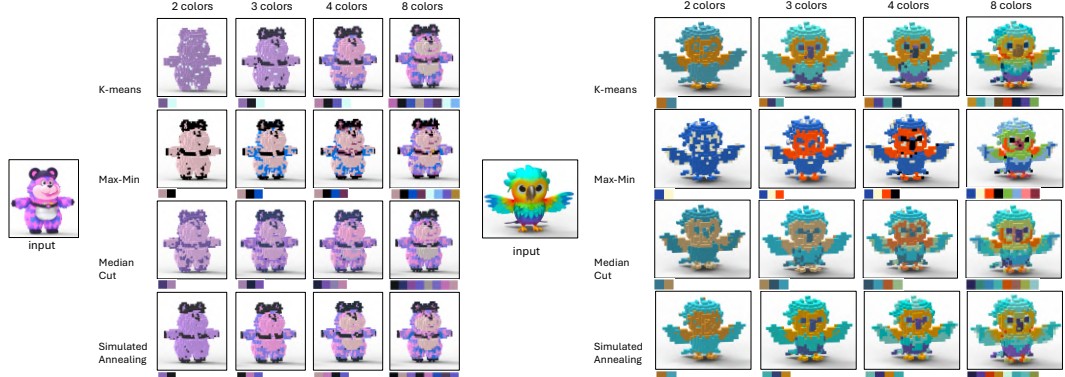

Figure 14: **Results with varying palette settings.** Examples using different palette extraction strategies and palette sizes.

**Results under different voxel sizes.** Fig. 15 illustrates voxel art renderings generated with varying voxel resolutions, demonstrating how grid granularity influences the level of abstraction, sharpness of edges, and overall visual fidelity of the outputs.

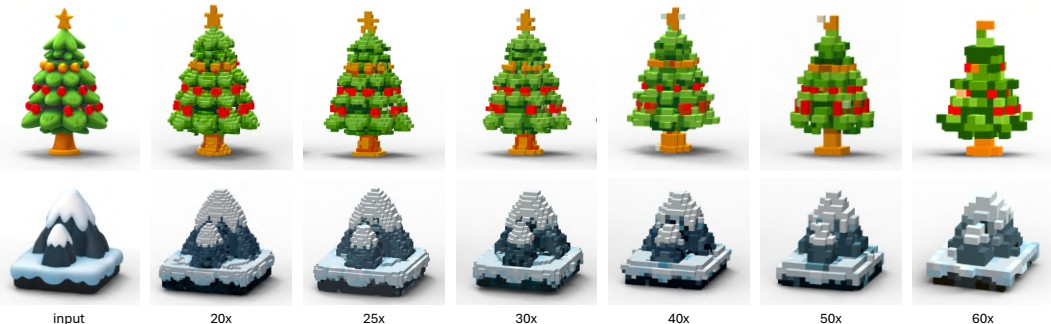

Figure 15: **Results under different voxel sizes.**

**Failure cases and analysis.** Finally, representative failure cases are shown in Fig. 16, mainly arising from complex shapes that exceed the capacity of the limited voxel resolution. These examples suggest that voxel art is better suited for capturing abstract details conveyed through color patterns and tonal contrasts, whereas fine-grained geometric intricacies are more likely to be lost under coarse discretization.

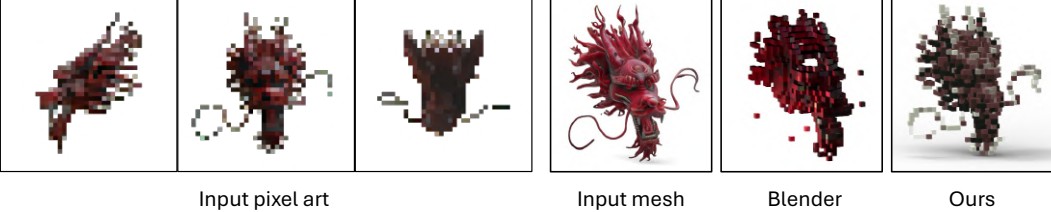

Figure 16: **Representative failure cases.**

**Comparison with single-image 3D reconstruction.** We also compare with **Rodin** Wang et al. (2022b), which performs well for image-to-mesh generation but is not designed for voxel art. As shown in Fig. 17, Rodin sometimes produces non-voxel outputs (right), and due to the single-image input, it often fails to capture reliable depth, resulting in flat structures (left). This further underscores the benefit of our multi-view voxel optimization pipeline.

Figure 17: **Comparison with Rodin** Wang et al. (2022b). Rodin excels at image-to-mesh but is not tailored for voxel art, often yielding non-voxel outputs (right) or flat geometry (left).

## A.4 USE OF LARGE LANGUAGE MODELS.

Large language models (LLMs) were used as general-purpose writing and editing assistants, such as improving the grammar, clarity, and readability of the text. The authors take full responsibility for the content of this paper.

