# OpenReview forum: "Voxify3D: From Mesh to Voxel Art with Palette Discretization and Semantic Guidance"
_ICLR.cc/2026/Conference — ICLR 2026 Conference Withdrawn Submission_

### Official Review · Reviewer_xC5N · 2025-10-26

**Soundness:** 3
**Presentation:** 2
**Contribution:** 3
**Rating:** 4
**Confidence:** 5

**Summary:**

This paper has proposed a mesh to voxel art pipeline. Basically, Voxify3D converts 3D meshes into stylized voxel art with discrete color palettes, pixel art supervision, and voxlel-based nerf. Extensive experiments have demonstrated the effectiveness of the proposed method.

**Strengths:**

1. The visual quality is very good.
2. The proposed method is straightforward can be finished in 2 hours for converting a single instance to voxels.
3. The evaluation is very comprehensive, both qualitatively, quantitatively, and also involves user study.

**Weaknesses:**

1. Overall, it is established on top of existing techniques (DVGO, , Pixel art generator, and clip-based semantic guidance). The discrete color selection / gumbel softmax part looks novel, but is also pretty straightforward.
2. The overall pipeline is still 2-stage tedious optimization, with many loss terms to tune (eq.11)
3. This method should be applicable to 3D scene right (like the furniture in 3d-front dataset), but currently it is working specifically on the 3d object.

**Questions:**

1. My main concern is whether this two-stage pipeline is right in the long term. Like in GaussianAnything (ICLR 25), the Fig. 4 has shown some feed-forward generated 3D asset with voxelized style (a voxelized dog). The trellis3d has also shown something similar in their project. The comparisons / discussions should be included for demonstrate that this two-stage pipeline is still the right solution in the current term.
2. I cannot see Fig. 3 (and tried it on two laptops / browsers), the writing needs to be improved.

---

### Official Review · Reviewer_AUMC · 2025-10-30

**Soundness:** 2
**Presentation:** 1
**Contribution:** 2
**Rating:** 2
**Confidence:** 4

**Summary:**

The paper proposes Voxify3D, a differentiable two-stage framework for transforming 3D meshes into semantically coherent stylized voxel art. The authors systematically evaluate various color palette extraction strategies and analyze their impact on stylization fidelity under different palette sizes. The key contribution encompasses the orthographic supervision strategy for precise pixel-voxel alignmentand flexible parameterization of palette size and selection methods to support diverse abstraction levels. The paper also identifies limitations in handling complex geometries under coarse discretization, offering insights into the trade-off between abstraction and detail preservation. Comparative experiments against state-of-the-art methods highlight the proposed approach's superiority in achieving both semantic alignment and pixel-art aesthetics. Overall, the paper provides a robust framework for voxel art synthesis, advancing the understanding of resolution-color-semantic interplay in discrete 3D stylization.

**Strengths:**

1. The implementation of color palette extraction strategies is methodologically sound, and the analysis of resolution-color-semantic trade-offs is theoretically justified.

2. The comparison against state-of-the-art methods is relevant and appropriately framed.

3. The work advances the understanding of discrete 3D stylization by explicitly modeling the interplay between resolution, color abstraction, and geometric fidelity.

4. The proposed framework demonstrates practical utility in achieving pixel-art aesthetics while maintaining semantic alignment.

**Weaknesses:**

1. The paper lacks a method diagram, making the pipeline’s structure and interactions between components unclear.

2. The related work lacks thematic categorization, presenting prior works as an unstructured list without comparative or contextual analysis.

3. The experiments focus on color-pattern-driven abstraction, but the method’s performance on tasks requiring fine geometric reconstruction (e.g., architectural modeling) is not evaluated, raising questions about broader applicability.

4. The failure cases highlight challenges in preserving intricate details under coarse discretization, yet the paper lacks a deeper discussion on potential solutions or architectural modifications to address these issues.

**Questions:**

1. The paper focuses exclusively on voxelization for pixel-art stylization as a downstream task. However, this application is highly domain-specific and relies on strong assumptions about color abstraction and geometric simplification. Could the authors elaborate on how their methodology could be adapted or generalized to other tasks where discrete volumetric representations are useful but require different optimization objectives? What components of their pipeline are transferable beyond the scope of artistic stylization?

2. The paper argues that PCG methods lack a unified optimization framework and require "extensive manual tuning." However, PCG’s core strength lies in its algorithmic flexibility and rule-based adaptability, which allows for dynamic content generation without explicit optimization. Does the proposed voxelization framework truly address these limitations, or does it merely shift the burden of manual parameter tuning to the discretization stage? How does the paper’s claim about "unified optimization" account for PCG’s inherent ability to generate diverse outputs through procedural rules rather than fixed optimization targets?

---

### Official Review · Reviewer_4EYk · 2025-11-01

**Soundness:** 3
**Presentation:** 2
**Contribution:** 3
**Rating:** 6
**Confidence:** 4

**Summary:**

This work proposes Voxify3D, a method for producing voxel abstractions of meshes with fixed sized color palettes. The approach consists of a two stage pipeline: stage 1 builds a coarse voxel representation to provide a base geometry and color while stage 2 refines this voxel representation with pixel-art images using a CLIP-based loss for semantic alignment and a depth-based loss to preserve geometry. The initial coarse stage uses DVGO to extract density and color values at each voxel. Then 6 orthographic views are rendered and compared to pixel art 2D generations to refine voxel density and color, both directly and with a semantic patch-based loss. To learn discrete colors, a pallet of a pre-defined number of colors is extracted from the generated images. Each voxel learns logits for each of the colors in this pallet which is then discredited by taking a Gumbel softmax of the logits. The overall method is evaluated both qualitatively through comparison figures and quantitatively through a CLIP-based metric and a user study. The submission also contains ablations of key method components.

**Strengths:**

- The proposed method provides an effective approach for abstracting meshes into fixed resolution and color voxel representations and outperforms existing baselines both qualitatively (Fig. 5) and quantitatively using a CLIP metric and user study.
- The technique for differentiably supervising the discrete colors of each voxel by using a Gumbel softmax on color logits is original, works well, and could be useful for future research on discrete pallet generation / stylization.
- The use of orthographic rendering to preserve voxel and pixel alignment is interesting.

**Weaknesses:**

- The presentation seems a bit rushed in places, for example Fig. 3 is captioned but does not actually appear in the submission.
- In some figures that show qualitative comparisons (i.e. Fig. 2 and Fig. 7), this paper opts to compare different approaches individually to Voxify3D on different examples for each approach. This makes it difficult to know if the comparisons shown hold for most cases or just on the single example displayed for each comparison. Showing figures where all approaches are compared on the same examples is more informative (as done in Fig. 5).

Overall, the results look promising, but the presentation could be more polished.

**Questions:**

- Have the authors considered using more complex clip-based semantic losses? 3Doodle [1] and Wir3d [2] find success using a semantic loss that combines standard CLIP embeddings, intermediate CLIP features from the unet, and LPIPS embeddings. I wonder if this type of semantic loss could improve results.
- Could the authors further clarify any differences between standard DVGO and the method used here to get the coarse voxel density and colors?

References:
[1] Choi, Changwoon, et al. "3doodle: Compact abstraction of objects with 3d strokes." ACM Transactions on Graphics (TOG) 43.4 (2024): 1-13.
[2] Liu, Richard, et al. "WIR3D: Visually-Informed and Geometry-Aware 3D Shape Abstraction." arXiv preprint arXiv:2505.04813 (2025).

---

### Official Review · Reviewer_ipqa · 2025-11-01

**Soundness:** 2
**Presentation:** 3
**Contribution:** 2
**Rating:** 4
**Confidence:** 5

**Summary:**

The paper presents Voxify3D, a differentiable two-stage framework for automatically converting 3D meshes into stylized voxel art. In the first stage, Voxify3D initializes a coarse voxel grid through voxel-based 3D reconstruction. In the second stage, Voxify3D refines the grid using six-view orthographic pixel-art supervision with discrete color palettes to achieve stylized, pixel-perfect results. Through experiments and user studies, Voxify3D demonstrates superior visual quality, semantic fidelity, and pixel-level aesthetics compared to existing voxelization methods.

**Strengths:**

S1. The storyline of this paper is clear and easy to follow.

S2. Automatically converting 3D meshes into stylized voxel art is an interesting topic.

S3. The proposed two-stage framework is reasonable.

**Weaknesses:**

W1. As shown in Figure 7, the proposed method includes a lot of components. I have doubts about whether all these components are necessary. First, as shown in Figure 7, there are only slight differences between using and not using the proposed component, such as pixel art, depth loss, and clip loss. Moreover, the paper compares with and without one specific component by different examples. I have no idea whether these results are cherry-picked or not. Actually, we need to conduct an ablation study of these components in a gradually increasing module manner to verify whether these components are necessary.

W2. The proposed method is too simple and straightforward, combining DVGO with some commonly used pixel and clip-based loss functions. I didn’t find any technique innovations. If I am wrong, please correct me in the rebuttal. I think the technological innovation of this paper has not reached the level of ICLR.

W3. Figure 3 can’t be displayed on my computer.

**Questions:**

Please see the weakness.

---

### Note · Authors · 2025-11-14

I have read and agree with the venue's withdrawal policy on behalf of myself and my co-authors.